# Non-equilibrium quantum domain reconfiguration dynamics in a two-dimensional electronic crystal and a quantum annealer

Jaka Vodeb [1,2,3] ✉, Michele Diego [1], Yevhenii Vaskivskyi [1,2], Leonard Logaric[1], Yaroslav Gerasimenko [1], Viktor Kabanov [1], Benjamin Lipovsek[4], Marko Topic [4] & Dragan Mihailovic [1,2,5] ✉

Relaxation dynamics of complex many-body quantum systems trapped into metastable states is a very active field of research from both the theoretical and experimental point of view with implications in a wide array of topics from macroscopic quantum tunnelling and nucleosynthesis to non-equilibrium superconductivity and energy-efficient memory devices. In this work, we investigate quantum domain reconfiguration dynamics in the electronic superlattice of a quantum material using time-resolved scanning tunneling microscopy and unveil a crossover from temperature to noisy quantum fluctuation dominated dynamics. The process is modeled using a programmable superconducting quantum annealer in which qubit interconnections correspond directly to the microscopic interactions between electrons in the quantum material. Crucially, the dynamics of both the experiment and quantum simulation is driven by spectrally similar pink noise. We find that the simulations reproduce the emergent time evolution and temperature dependence of the experimentally observed electronic domain dynamics.

Many-body systems emerging through symmetry-breaking phase transitions often end up in inhomogeneous intermediate states before reaching a homogeneous ground state. In the aftermath of second-order transitions, such as the superconducting transition in type-2 superconductors, fluctuations in the phase of the order parameter force causally unconnected regions to evolve independently, forming mesoscopic vortex structures[1–3]. In the aftermath of first-order phase transitions, the classical dynamics involves nucleation of the new phase, followed by coalescence[4] and "ripening"[5]. Topological and jamming transitions present more complex non-equilibrium dynamics that recently became topics of wider interest[6,7]. In classical systems, the kinetics is primarily diffusion-driven, while metastability arises from symmetry or topological constraints. In quantum systems[8], particularly at low temperatures we may expect to observe either coherent quantum dynamics for perfectly isolated systems, or incoherent macroscopic tunneling (IMT) aided by noise-assisted processes causing transitions between different mesoscopic state configurations[9].

The challenge is to quantitatively measure state reconfigurations of a many-body system in order to identify mesoscopic tunneling processes between different domain configurations, and then compare such data with model simulations of the dynamics. Because such processes are quite common, and have important potential

[1]Jozef Stefan Institute, Jamova 39, 1000 Ljubljana, Slovenia. [2]Faculty of Mathematics and Physics, University of Ljubljana, Jadranska 19, 1000 Ljubljana, Slovenia. [3]Institute for Advanced Simulation, Jülich Supercomputing Centre, Forschungszentrum Jülich, Wilhelm-Johnen-Straße, 52425 Jülich, Germany. [4]Faculty for Electrical Engineering, University of Ljubljana, Tržaška 25, 1000 Ljubljana, Slovenia. [5]CENN Nanocenter, Jamova 39, 1000 Ljubljana, Slovenia. ✉e-mail: jaka.vodeb@ijs.si; dragan.mihailovic@ijs.si

applications[10], the ability to faithfully simulate their dynamics on a quantum level would lead to significant progress in understanding metastable states in quantum materials and non-equilibrium quantum devices. In electronic crystals, the ordering of domains and discommensurations (domain-walls) are traditionally modeled classically by an anisotropic Ising model with competing interactions[11] or by continuum models[12,13], but these models give little insight into how inhomogeneities evolve in the quantum regime. Ab-initio methods can give detailed electronic band structure information of individual domain walls[14], but cannot be used to simulate the emergent many-body collective dynamics. Non-equilibrium dynamics pose an especially difficult challenge to faithful classical simulation[15], where an exponentially large Hilbert space dimension is required. Brute force state vector simulations can barely exceed 50 qubits even on the best high-performance computer clusters currently available[16]. On the other hand, quantum Monte Carlo calculations with added noise are not developed for direct probing of non-equilibrium dynamics, and it is still an open question whether, and how they could be applied at all. Quantum simulations of ground state properties in many body systems, as well as metastable decay of complex systems using ultracold atomic lattices have recently been demonstrated[17,18], and a lot of recent attention has focused on noisy intermediate scale quantum (NISQ) processors. But extension to real-world non-equilibrium systems requiring thousands of qubits in two dimensions, such as is addressed here, is particularly challenging because it requires extreme, currently unavailable computational resources, as well as presently still unachievable control of decoherence.

Macroscopic quantum motion of domain walls in the presence of noise has attracted a lot of attention since the 1980s. Typically, spin tunneling in a ferromagnets[19,20] was considered, which may be formulated in terms of domain wall tunneling[21,22]. Research of this type includes magnetotransport measurements on thin ferromagnetic wires[23], and magnetization experiments on single particles[24,25], nanomagnet ensembles[26-28] and rare-earth multilayers[29]. A different method involves investigating macroscopic disordered ferromagnets[30-33]. Here we report for the first time a study of quantum noise driven domain wall dynamics of strongly correlated electrons. Our experimental system is the prototype quasi-two-dimensional (2D) strongly correlated transition metal dichalcogenide material 1T-TaS$_2$[34], chosen because it displays prototypical electronic crystal ordering behavior[35]. As a result of the strong electron-phonon interaction in this material, the electronic kinetic energy $K$ is exponentially reduced[36], and Coulomb energy $V$ becomes dominant[37]. The essential physics in this regime thus comes from the formation of mutually repulsive heavily dressed quasiparticles−polarons[34], with a variety of polaron-ordered states and metastable domain structures[34,38,39]. In response to external perturbation 1T-TaS$_2$ exhibits fundamentally interesting and practically useful switching behavior[10,34,40-42] between different metastable states associated with different configurations of domains of the low-temperature electronic superlattice. At low temperatures, the temporal reconfigurations can be effectively investigated by scanning tunneling microscopy (STM). The classical many-body relaxation dynamics of such domain states is topologically inhibited[43,44], effectively creating an energy barrier for the transition between different domain configurations. In such a system where the dynamics of classical processes is topologically inhibited, quantum dynamics is necessary for metastable state relaxation, especially at low temperatures. Understanding these processes is directly relevant for controlling functional properties, and in particular data retention in non-volatile configuration memory devices[10,45].

Using a new approach in modeling emergent non-equilibrium behavior, we investigate the electronic domain dynamics in 1T-TaS$_2$, with a parallel simulation using a superconducting quantum simulator. The decoherence of domains in the simulation−just as in 1T-TaS$_2$−is governed by $1/\nu$ pink noise[46,47], so the simulations can be thought to utilize the principles of simulating nature with nature, as originally proposed by Feynman[48]. Here we find that the quantum simulator describes the natural noise-driven processes remarkably well. The reproduction of the emergent metastable decay behavior that is observed in the experiments is particularly striking. The results lead to valuable insights into how microscopic many-body interactions between electrons on a superlattice conspire to display metastable quantum domain relaxation dynamics observed in the experiments on 1T-TaS$_2$.

## Results

### Domain reconfiguration measurements in 1T-TaS$_2$

In our experiments, a metastable domain state in the $\sqrt{13} \times \sqrt{13}$ electronic superlattice of 1T-TaS$_2$ is set up by electrical pulse charge injection through an STM tip[43,49,50] (Fig. 1c). As the domain structure evolves in time, its configuration is recorded periodically by STM at different temperatures. A relaxation sequence at 5 K is shown in Fig. 1c. The tip is retracted and set to zero bias between measurements to avoid reconfiguration by spurious signals from the tip. This measurement results in classical sequential "snapshots" of the domain configurations at different temperatures (see Supplementary Notes 1 for experimental details). The reconfigurations between sequential images are quantified by measuring the Hamming distance, defined here as the number of altered occupied polaron positions between frames (Fig. 1c), and expressed as the fraction of polarons moved $f = \Delta N/N$, where $N$ is the total number of polarons per frame (Fig. 1d). Characteristically, jumps of $f$ appear in time (Fig. 1d), reflecting the discrete nature of the domain reconfiguration process[51]. When $f$ is averaged over a large number of scans on different areas of the sample an approximately exponential decay is observed (Fig. 1e). The classical power law time dependencies $f(t) \sim r(t)^2 \sim t^{-1}$ or $t^{-2/3}$, that would be characteristic of nucleation and coalescence processes[4], respectively, at higher temperatures[52], cannot be made to fit the 5 K domain relaxation data (Fig. 1e). Considering the difference of the processes involved, this is not surprising.

A crucial test of the experiments is to see if, and how, the STM tip influences the reconfiguration process. Comparing $f(t)$ obtained by tip scanning at regular intervals (Fig. 1e), with $f(t)$ measured under identical experimental conditions, but with a retracted tip over a longer time interval, we see in Fig. 1f that the two relaxation times are very similar: $\tau = 1049 \pm 90$ s and $943 \pm 80$ s, respectively, implying that within experimental error the influence of the STM tip is negligible. A number of other tests that support this conclusion are given in the Supplementary Notes 1, including a detailed calculation of Joule heating by the STM tip current (see Supplementary Notes 1.2). We have thus established that the noise in 1T-TaS$_2$ does not originate from the STM tip, but has a different non-thermal origin. Other possible excitation sources are acoustic phonons, quantum fluctuations in two-level systems (TLSs) in the sample, the substrate and at the interface, as well as spurious external electro-magnetic fields brought by the grounded sample contact. Importantly for our purposes, in 1T-TaS$_2$ the noise has been experimentally shown to have a $\sim 1/\nu$, frequency dependence[46].

The dependence of the reconfiguration rate $R(T) = \frac{1}{\Delta t} \int_0^{t_{\exp}} f(t) dt$, where $\Delta t$ is the time interval between subsequent image samplings and $t_{\exp}$ the duration of the experiment at temperature $T$, on sample temperature in 1T-TaS$_2$ is shown in Fig. 1g for $\Delta t \simeq 480$ s. $R(T)$ exhibits a distinct crossover from $T$-independent to $T$-dependent behavior at a temperature $T_0 \approx 20$ K. The data can be fit well to: $R(T) = R_q + R_0 \exp(-E_M/k_B T)$, where $R_q \simeq 8 \pm 1 \times 10^{-4}$ s$^{-1}$ is the $T$-independent tunneling rate, and the second term describes $T$-activated hopping across a barrier. The value $E_M = 6 \pm 3$ meV obtained from the fit to the data in Fig. 1g is not far from the activation energy $E_A = 10 \sim 20$ meV obtained from previous macroscopic resistance relaxation measurements of bulk samples[51], providing an important experimental consistency check.

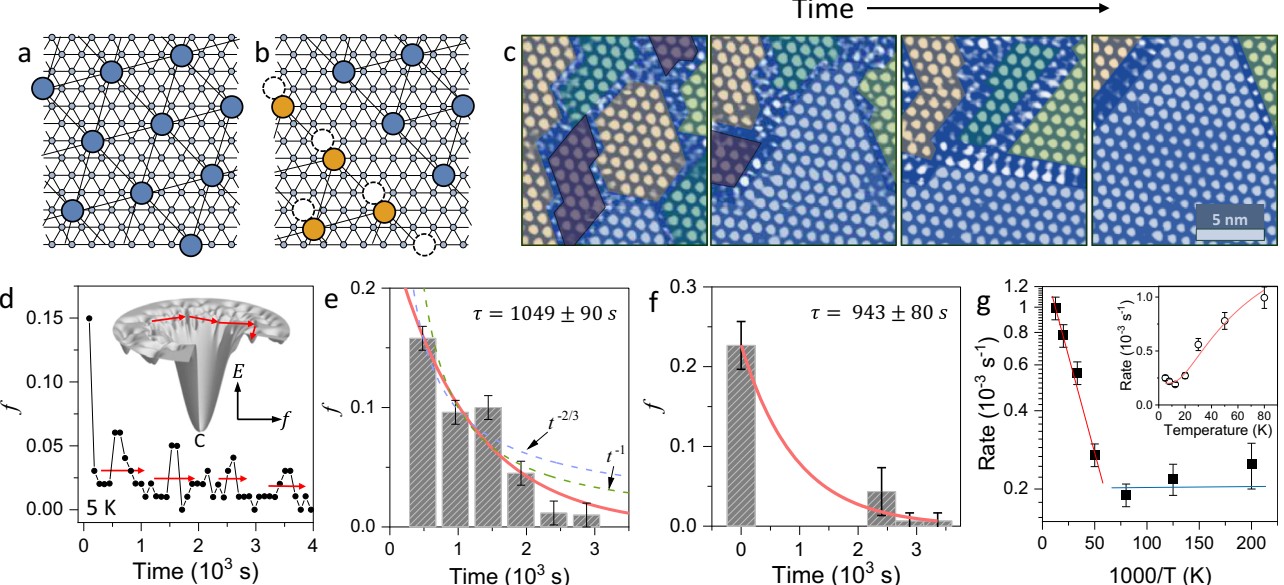

**Fig. 1 | Domain reconfigurations in 1T-TaS2. a** The crystal lattice showing Ta atoms (small circles). Occupied electron sites (blue and orange circles) **a** in the ground state, and **b** near a domain wall. **c** A series of STM measurements at regular time intervals showing domain reconfiguration. Domains are colored to signify a displacement relative to the main commensurate lattice (light blue) (STM parameters: −0.8 V, 50 pA). **d** The fraction of electrons moved $f(t)$ as a function of time measured within the same area at 5 K. The jumps correspond to large (observable) configurational changes. The inset shows a schematic of the configurational free energy $E$ as a function of configurational coordinate (Hamming distance) $f$. The excited states within each well represent states which conserve $f$ (see text). **e** $f(t)$ deviation averaged over different areas of the sample, with periodic STM scans. Exponential and power law fits to the data are shown (solid and dashed lines, respectively). Error bars represent counting errors. **f** Same as (**e**), except with a 32-min gap in scanning the first point. The exponential fit is the red line. **g** The temperature dependence of $R$ averaged over a large number of decays on a logarithmic scale with the inset showing a linear scale. Counting errors in $f(t)$ are directly related to error bars in $R$.

Since by STM we observe "snapshots", which are effectively separated by large time windows, we assume that reconfigurations correspond to single escape events from a local minimum in a multidimensional free energy landscape of dimension $D = 2^N$, where $N$ is the number of atomic sites, shown schematically in 1D as a function of $f$ is in the insert to Fig. 1d. Different minima in $E(f)$ correspond to different configurational states, while the excited states within the wells correspond to configuration processes which conserve $f$, such as translations.

**Quantum decay rate**

First we check whether existing analytical theory of noise-mediated quantum dynamics in the presence of a thermal bath can explain the crossover in $R(T)$. The canonical Kramers-Caldeira-Legget (KCL) quantum decay rate theory[53,54] assumes a system moving through a cubic potential in a 1D configurational landscape and does indeed predict an $R(T)$ crossover. Here, we approximate the movement of polarons in 1T-TaS2 through its multidimensional energy landscape with a series of local 1D paths, each representing a single reconfiguration event (see insert to Fig. 1d). According to KCL, polarons in 1T-TaS2 are initially trapped in the metastable minimum of the cubic potential separated by a barrier from a lower energy state, which can be overcome by either noise-mediated incoherent quantum tunneling at low $T$ or thermal activation at high $T$. The crossover occurs at temperature $T_0$ within a range $\Delta T$. If we take the experimental values $E_M$, $T_0 = 20$ K and $\Delta T = 10$ K, obtained from Fig. 1g and plug them into KCL theory in order to characterize an average reconfiguration event, it predicts $\frac{E_M}{\hbar\omega_0} \simeq 0.4 \sim 1.4$, where $\hbar\omega_0$ is the kinetic energy scale of polarons in 1T-TaS2. This is in direct contradiction with the underlying assumptions of KCL theory $\left(\frac{E_M}{\hbar\omega_0} \gg 1\right)$. Therefore, the assumptions of the KCL theory are fundamentally incompatible with the observed behavior in 1T-TaS2. An extension of the KCL theory into the multidimensional case as well as $E_M \sim \hbar\omega_0$ is required, which unfortunately currently does not exist.

**Quantum simulation of domain reconfiguration dynamics**

An alternative to analytical theory is simulation of the system microscopically, and examine the emergent behavior in comparison with experiments of the dynamics of domains in 1T-TaS2. We start with a single Ta band of tight-binding electrons[55] coupled to phonons via the Fröhlich interaction, interacting via a long-range Coulomb repulsion $V_c(i,j)$. In the limit of strong electron-phonon interaction, such a model exhibits repulsive polarons as slow quasiparticles with an exponentially suppressed tunneling matrix element $t_{ij}$[36]. Here we focus on charge ordering of such polarons and neglect the spin degree of freedom, attributing to each atomic site a binary variable $q_i \in \{0,1\}$ (a qubit), which represents the polaron occupancy of site $i$ on a triangular lattice and assume only nearest neighbor repulsion $V(i,j)$. The Hamiltonian is then $H_M = -\sum_{\langle ij \rangle} t_{ij} c_i^\dagger c_j + \sum_{\langle ij \rangle} V(i,j) q_i q_j - \mu \sum_i q_i$, with $c_i^\dagger c_i = q_i$, and chemical potential is $\mu$, which is chosen such that the ground state at $t_{ij} = 0$ is a triangular polaronic superlattice where each polaron can occupy one of 3 atomic lattice sites (Fig. 2a).

Quantum dynamics calculations of sufficient system sizes required to capture reconfiguration dynamics of hundreds of polarons in the experiment are presently unrealistic on a classical digital computer. We thus explore the alternative possibility of using a programmable quantum annealer D-wave Advantage 6.1 (PQA) to model the dynamics, which has already been shown to exhibit multiqubit tunneling in the presence of thermal noise[56–59]. Importantly, the superconducting qubit TLS dynamics of the PQA is mediated by coupling to TLSs in the environment which has a very similar $1/\nu$ frequency dependence of the noise as 1T-TaS2[46,47]. (A more detailed comparison of noise characteristics is made in Supplementary Notes 2.5) The PQA is designed for simulations that are extensions of the transverse field

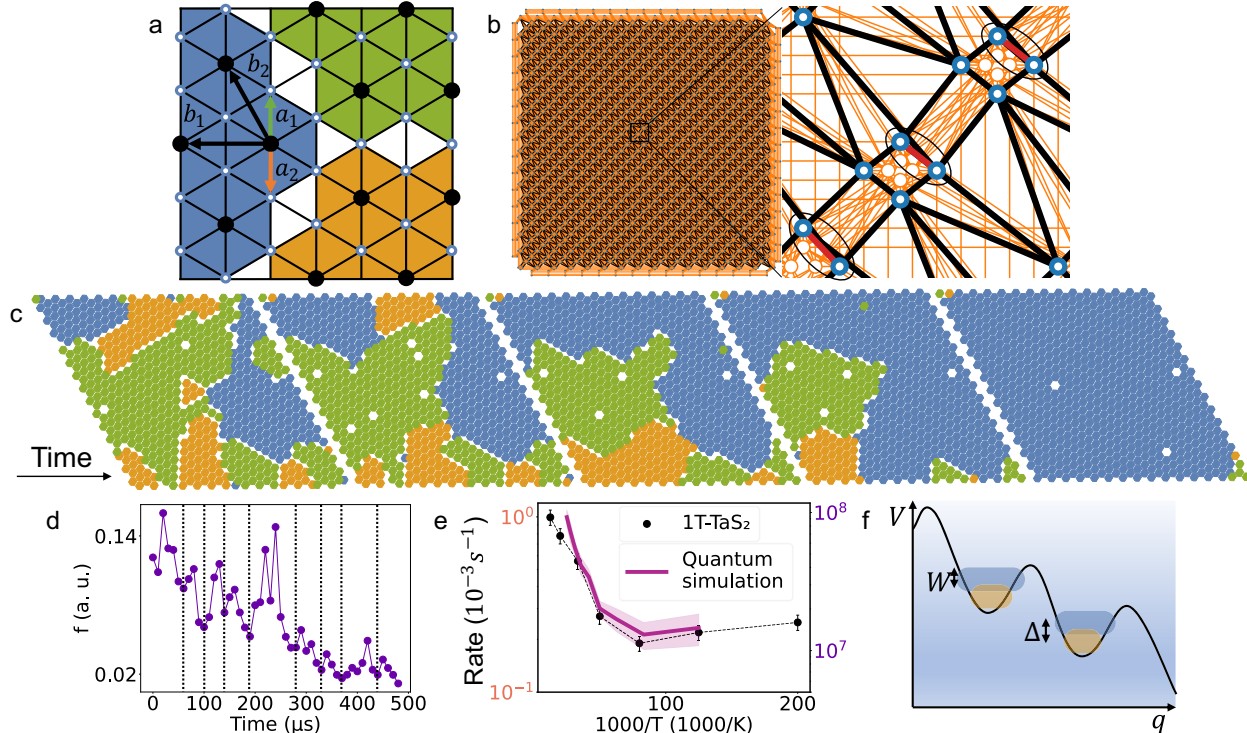

**Fig. 2 | Simulation of emergent quantum decay based on microscopic quantum simulations. a** The three possible ground states of the nearest neighbor repulsion we deployed on the quantum annealer. They are a 1/3 polaronic lattice with primitive vectors $b_1$ and $b_2$, shifted by two possible atomic lattice vectors $a_1$ and $a_2$. The different superlattices are colored for clarity. **b** Embedding of the triangular lattice used in our simulations on the PQA with the inset showing the detailed connections between qubits. The white dots encircled with blue represent individual atomic sites (also in (**a**)), black lines are connections between qubits we used for the nearest neighbor repulsion, red lines are physical connections used for creating logical qubits (also encircled in black) forming a triangular lattice and the orange lines are the unused physical connections on the processor. **c** An example of

a relaxation sequence with parameters $t_a = 5\,\mu s$, $r = 9$ and $T_{eff} = 1.33$. **d** The Hamming distance $f(t)$ sampled from the quantum annealer for the sequence shown in (**c**). This can be compared with experimental data in Fig. 1d. Black dashed lines show the intermediate metastable states that emerge during the relaxation process. **e** $R(T)$ extracted from the PQA with parameters $t_a = 5\,\mu s$ and $r = 15$, and 1T-TaS$_2$ with the shaded area and error bars representing the standard deviation when averaging across multiple relaxation measurements. The temperature scale for the PQA was chosen to be $T = T_{eff} * 4K$, for demonstration purposes. See Supplementary Notes 4.3 for actual parameter estimation. **f** The washboard potential of the PQA flux qubit system with the spacing between 2 levels in the system $\Delta$ coupled to the external bath with an integral over noise $W$.

Ising model (TFIM) with adjustable temperature:

$$H_{PQA}/k_B T = \left( -\sum_i \sigma_i^x/2 + r[s(t)]\left( \sum_{\langle ij \rangle} V(i,j)q_i q_j - \mu \sum_i q_i \right) \right) / T_{eff}[s(t)] \quad (1)$$

where $\sigma_i^z = 2q_i - 1$, $\sigma_i^{x,z}$ are Pauli matrices operating on $q_i$ and $s(t)$ is a custom time-dependent function ranging from 0 to 1, conventionally called the annealing schedule in the simulation. The TFIM model was originally introduced to describe collective motion between of protons in microscopic hydrogen-bonded double well potentials[60]. It is known that solutions of the TFIM in 1D can be used to model domain walls propagating as free particles in the background of strong interactions with an associated false vacuum state that describes the transitions between different domain configurations[61]. It has also been shown recently[62] that with a 2D TFIM with strong nearest neighbor interactions, domain walls behave as free fermions with conventional hopping $\sum_{\langle ij \rangle} c_i^\dagger c_j + h.c.$ in the limit of strong interactions ($r \gg 1$), which is closely analogous to domain walls between polaronic lattices. We observe similar domain wall dynamics in both 1T-TaS$_2$ and the PQA, suggesting that in the strong interaction limit the effective Hamiltonians are indeed the same. Through $s(t)$ we are able to control the only two tunable parameters in our simulation: the effective dimensionless temperature $T_{eff}[s(t)]$ and the ratio between the potential and kinetic energy, $r[s(t)]$[63]. We map $\mu$ onto individual qubits biases and $V(i,j)$ to couplers between qubits, with a 4/3 ratio of physical to

logical qubits (Fig. 2b). In this way, the TFIM makes it possible to describe the tunneling dynamics between different 2D domain states of 1T-TaS$_2$.

The calculation should be considered as a salient model that ignores coupling to phonons, as well as long-range interactions, but given that the classical charged lattice gas modeling describes low-temperature ordering in 1T-TaS$_2$[38], the use of the TFIM in the limit of strong interaction may be justified for weak quantum tunneling effects at low $T$. The simulation does not explicitly address the sources of environmental noise[64], but it is assumed to have $1/\nu$ frequency dependence, similar as in 1T-TaS$_2$[46], based on previous measurements of the PQA[47]. (More discussions on the application of the TFIM model on the D-Wave Advantage quantum annealer is given in Supplementary Notes 3).

We emulate the STM measuring process by initializing the system of qubits (polarons) into a domain state (as a product state in the qubit $q_i$ basis) similar to the initially observed states in 1T-TaS$_2$. The domain walls separate the 3 possible ground states. (For simplicity, in the simulation we have used 3 rather than the 13 possible degenerate ground states in 1T-TaS$_2$). The PQA allows us to adiabatically tune $r[s(t)]$ and $T_{eff}[s(t)]$ via $s(t)$ in order to bring them to the range of values for which we want to observe dynamics. After we let the system of qubits evolve for a set amount of time, we perform a measurement and observe the polaronic configuration that the qubits have evolved into. Then, we use the measured configuration as the new initial product state, thereby faithfully emulating the iterative

STM measurement. From the point of view of Hamiltonians, we start with $H_{PQA}(t=0) = (-\sum_i \sigma_i^x + r[1](\sum_{\langle ij \rangle} V(i,j)q_i q_j - \mu \sum_i q_i))/T_{eff}[1]$, where $r[1] \approx 10^{10}$, and initialize the qubits into the initial domain state in the qubit computational basis, then adiabatically tune to $H_{PQA}(t=\frac{t_a}{2}) = (-\sum_i \sigma_i^x/2 + r[s(\frac{t_a}{2})](\sum_{\langle ij \rangle} V(i,j)q_i q_j - \mu \sum_i q_i))/T_{eff}[s(\frac{t_a}{2})]$ and let the system evolve in time, and finally adiabatically tune back to $H_{PQA}(t=t_a) = H_{PQA}(t=0)$, which constitutes a measurement. The annealing time $t_a$ is of the order of 10 μs and the energy scale of $H_{PQA}$ is a few GHz. We focus on values $r[s(\frac{t_a}{2})] \gg 1$, where the kinetic energy of domain walls is small compared to the interaction energy and study the behavior of domain wall dynamics at different $T_{eff}$. Details on the methodology are given in the Supplementary Notes 4.

We observe translations and melting of the domain walls towards one of the 3 uniform ground states from the initial domain wall state, just as it occurs in 1T-TaS$_2$ (Figs. 1c and 2c). The characteristics of the domain melting processes captured by the fraction of moved polarons $f(t)$ in 1T-TaS$_2$ (Fig. 1d) also experiences intermediate saturations and jumps in the PQA (Fig. 2d). Most importantly, both systems exhibit a crossover in the relaxation rate $R(T_{eff})$ from $T$-dependent activation processes at high $T$ to $T$-independent noise-mediated quantum tunneling events at low $T$ (Figs. 1g and 2e). $R(T_{eff})$ saturates to a $T$-independent non-zero rate due to slow system dynamics compared to the sampling time, which is the rate at which we measure with the STM tip in 1T-TaS$_2$ or the rate of qubit measurements in the PQA. If the system is given enough time to fully relax to the uniform ground state, then the dynamics eventually stop. In order to reproduce the experimental timescale of the dynamics ($\sim 1000$ s), we need to set $r \approx 58$. As an indirect comparison, the usual criterion for Wigner crystallization in 2D is $r_s = \frac{e^2 m^*}{\hbar^2 \sqrt{n}} = 31\text{--}38^{35,65}$, justifying the use of a dominant interaction term compared to a small kinetic energy term in the quantum simulation.

## Discussion

Let us interpret the physical processes involved in noise-mediated quantum tunneling. 1T-TaS$_2$ and the PQA are both described by a Hamiltonian $H_S$, coupled via $H_I$ to an external thermal bath B with a $1/\nu$ noise spectrum at temperature $T$, where the total Hamiltonian is $H = H_S + H_I + H_B$. We can represent an arbitrary bath described by a spectral function $\Im(\omega) = \frac{\pi}{2} \sum_\alpha (C_\alpha^2/m_\alpha \omega_\alpha) \delta(\omega - \omega_\alpha)$ with a system of decoupled harmonic oscillators (HOs) with mass $m_\alpha$, frequencies $\omega_\alpha$, and couplings $C_\alpha$ in $H_I^9$. Temperature effects are taken into account by populating each HO with a Bose-Einstein distribution with temperature $T$, meaning that at low $T$, HOs effectively reduce to two-level systems. In the limit of $H_I \rightarrow 0$, we would only observe pure tunneling events from an initial classical polaronic configuration into other configurations, determined by their overlap with instantaneous eigenstates during the adiabatic evolution as well as the shape of the annealing schedule. By bringing the initial classical configuration adiabatically into a superposition state of multiple configurations and then back, we introduce a probability of tunneling to other configurations involved in the superpositions. We found that the kinetic energy contribution is very small compared to the interaction ($r \approx 58$), which means that pure tunneling events would be highly constrained in the energy difference between the initial and final configuration. If fact, the energy difference tends to 0 in the limit of $r \rightarrow \infty$, meaning we would have no configurational changes at $T = 0$ and zero noise.

Now we discuss the effect of noise and temperature on quantum tunneling. The number of HOs, populated at temperature $T$, required to describe a particular spectral function is much larger than the number of polarons, meaning that they effectively serve as an infinite energy reservoir for the polarons and are constantly bringing them into thermal equilibrium with respect to $T$. Noise is represented by the influence of HOs on the energy spectrum of $H_S$, which is a static energy shift of energy levels$^{64}$. The simplest example is when both $H_B$ and $H_S$ are two-level systems. The two

levels of $H_S$ represent two polaronic eigenstates and the two levels of $H_B$ are the first two HO levels. Each polaronic state is split by $\hbar \omega_\alpha$ into two levels by adding the HO degree of freedom, meaning that a properly chosen $\omega_\alpha$ can bring two polaronic states to the same energy, thereby making quantum tunneling between them possible. This is what we mean by noise-mediated macroscopic quantum tunneling. We introduce the $1/\nu$ noise spectrum by increasing the number of HOs and choosing appropriate $C_\alpha, m_\alpha$ and $\omega_\alpha$. The polaronic state energy level then becomes an energy band with width $W$, given by an integral over the noise spectrum, as shown in Fig. 2f. If energy bands overlap, then a transition between the bath degrees of freedom can already induce a polaronic configurational change. If they do not overlap, the energy gap is still reduced by $W$, increasing the probability for a transition. Both represent incoherent macroscopic tunneling (IMT) processes.

Quantum simulation gives us some other unique insights into the emergent tunneling phenomenon. The dynamics of the two systems is governed by an anharmonic metastable state potential in terms of the configurational coordinate, measured by $f$, for 1T-TaS$_2$ and the PQA respectively. While the washboard potential and two-level system dynamics of flux qubits in the PQA emerge from the device concept, the energy landscape of the metastable state of 1T-TaS$_2$ is determined by the topology and the microscopic details of the domain structure arising from discommensurations$^{14}$. The extracted parameters from the simulations describing the emergent processes are the barrier energy $E_B$ (height) and the barrier width $w$ which describe the rate of tunneling between domain configurations. In 1T-TaS$_2$, the thermal activation energy $E_B$ is determined directly from the $T$-dependence of the rate in Fig. 1g or resistivity relaxation to be $E_M \simeq 10 \sim 20$ meV. On the other hand, $w$ is related to $f$, where both are measured in terms of the configurational coordinate, which is set by the state of $N$ Bloch spheres representing qubits, parametrized by $2N$ spherical angles. The plot of $f(t)$ in Fig. 1d thus reflects the tunnel barrier width for each measured reconfiguration step.

The motivation for the present quantum treatment arose from the fact that temperature-independent electronic domain relaxation in 1T-TaS$_2$ at low temperatures cannot be explained by classical processes because their relaxation is topologically inhibited$^{42}$. Yet we find that the phenomenological parameters extracted from the $T$-dependence of the relaxation rate and the crossover to $T$-dependent dynamics are inconsistent with conventional quantum decay. Introducing a new approach for modeling the domain dynamics we use a microscopic model implemented on a superconducting qubit processor, directly mapping the electron-electron interactions onto the PQA inter-qubit connections. The crucial next step demonstrated here is to treat the noise from the surroundings as coupled two-level systems (HOs) with the ubiquitous $1/\nu$ frequency dependence that has been shown experimentally to be present in both 1T-TaS$_2^{46}$ and the PQA$^{47}$. The emergent many-body dynamics and associated domain reconfigurations are thus simulated solely on the basis of microscopic interactions with a single adjustable parameter $r$ tuning the 'quantum-ness' of the system. The simulations clearly display the crossover from $T$-activated dynamics to $T$-independent domain melting that is experimentally observed in the STM measurements, giving direct insight into how the phenomenological relaxation rates emerge from microscopic interactions. The present calculations demonstrate the usefulness of simulations on a noisy quantum processor for modeling emergent non-equilibrium many-body dynamics. With appropriate mapping, such an approach can be applied to other non-equilibrium open quantum systems that are subject to external sources of incoherent noise—including quantum materials such as the quantum paraelectric SrTiO$_3^{66}$, various hydrogen-bonded systems$^{67}$, in the folding of proteins through a similarly multidimensional, topologically defined energy landscape$^{68}$, or tunneling between false vacuum states that emerge from elementary interactions$^{69-71}$. Apart from fundamental

interest, modeling quantum reconfiguration dynamics is important for low-temperature memory devices[41], in which quantum domain melting processes functionally limit the long-term data retention[51].

## Reporting summary

Further information on research design is available in the Nature Portfolio Reporting Summary linked to this article.

## Data availability

STM image and measurement data from the quantum annealer are available in Supplementary Information.

## Code availability

Code is available on GitHub[72].

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

## Acknowledgements
We wish to acknowledge discussions with Tomaž Prosen, Marko Žnidarič, Andrew King, and Tomaž Mertelj. Single crystals used in this work were grown by Petra Šutar. This project has received funding from ARIS programmes/project P1-0040 (J.V., M.D., Y.V., L.L., V.K., and D.M.), P2-0415 (B.L. and M.T.), N1-0092 (Y.V., Y.G., and D.M.), young researcher grant P08333 (J.V.), and from the European Union's Horizon 2020 research and innovation program under the Marie Skłodowska-Curie grant agreement No 701647. J.V. acknowledges support from the project Jülich UNified Infrastructure for Quantum computing (JUNIQ) that has received funding from the German Federal Ministry of Education and Research (BMBF) and the Ministry of Culture and Science of the State of North Rhine-Westphalia.

## Author contributions
M.D., Y.V., Y.G., and D.M. performed experimental measurements and analysis of experimental data. J.V. conceptualized and performed quantum simulations on D-Wave's quantum annealer. L.L. and V.K. performed theoretical analysis of data from experiment and simulation. B.L. and M.T. performed simulations of Joule heating from the STM tip. J.V. and D.M. wrote the paper. All authors contributed to the supplementary information.

## Competing interests
The authors declare no competing interests.
