## [Peer Review File · Nature Communications]

REVIEWER COMMENTS

Reviewer #1 (Remarks to the Author):

In this manuscript, Jaka Vodeb et al. report a combined study of STM and quantum simulations about the time-evolution of the domain reconfiguration in 1T-TaS₂. The authors first prepare the domain structure in 1T-TaS₂ by the electrical pulse charge injection through STM tip. They record the domain structure periodically by STM and analyze the Hamming distance as a function of time. The authors further use the programmable noisy superconducting quantum simulator to simulate the reconfiguration dynamics. I am not convinced that simulating the domain reconfiguration dynamics in 1T-TaS₂ is important. I think this manuscript does not meet the high standard of Nature Communications. I do not recommend the publication of this manuscript in Nature Communications. Below are my detailed comments.

- (1) What is the time interval for taking the STM topographies in Fig. 1c? Are the STM topographies in Fig. 1c taken on the same area?
- (2) Since the author also report the temperature-dependent reconfiguration dynamics, in the main text, they should also show similar series of STM topographies as in Fig. 1c, but measured at different experimental temperatures.
- (3) For the fixed temperature, does the reconfiguration rate change for difference areas?
- (4) In comparison with Fig. 1e, there are only a few data points in Fig. 1f. In order to make better comparison, the authors should measure more data points with the STM tip retracted.
- (5) The authors perform the experiment with STM tip retracted to confirm that the noise in M does not originate from the STM tip. But it could originate from the electric field of the STM tip, because the electric field is long ranged. The authors should try to retract the tip, and reduce the bias voltage at the same time.
- (6) Why the authors do not fit the last data point in Fig. 2g?
- (7) Since they are several parameters in the simulation, I am not surprised to see that the authors can more or less fit the reconfiguration rate. The authors should provide stronger evidence that their quantum simulation is indeed applicable to the domain reconfiguration dynamics in 1T-TaS₂.

Reviewer #2 (Remarks to the Author):

The manuscript by Vodeb et al. reports a joint experimental and theoretical work on the relaxation dynamics of macroscopic domains in a 2d electronic crystal and its modeling using a quantum annealer machine.

The relaxation dynamics of complex many-body or quantum field systems brought out of equilibrium and trapped into metastable states is presently a very active field of research from both the theoretical and experimental point of view. This is a very interdisciplinary field of research that connects concepts of condensed matter physics and quantum dynamics and has deep implications even in cosmology for the so-called 'false vacuum decay'.

The present work specifically attacks the phenomenology of domain wall relaxation: in contrast to the traditional cosmological false vacuum decay where the decay process connects two spatially-homogeneous configurations differing by some macroscopic observable (aka two vacuum states), here the decay process consists of the motion of domain walls spatially separating different crystalline arrangements. As a common feature of the two phenomena, the temporal dynamics involves overtaking some potential energy barrier to exit from a metastable local minimum of the energy landscape. Whereas the physics of domain wall relaxation is -at least to my eyes- less intriguing and challenging than the traditional false vacuum decay, still the topic of the manuscript can be of some general interest for a wide community of physicists. In any case, I strongly recommend the authors to update the title of the manuscript and several parts of the text to faithfully match the actual content of the work and avoid misleading the readers.

As a most interesting experimental observation, the dynamics of a complex, density-modulated electronic state is followed in real time by taking successive STM snapshots of the temporal evolution of the electron configuration after the system has been initialized in a domain state with a suitable current pulse. The main observation is that the relaxation follows an exponential law and its rate tends to a constant value at low temperatures and follows an thermally-activated-like behaviour at higher temperatures. No study of the dependence of the rate on other physical parameters is provided and only a generic interpretation of the activation energy value is provided. Such thermal activation behaviours are well-known in the literature on domain walls; also the specific material system under investigation, its electronic density-modulated states, and its STM imaging are well-established in the community, so they can be hardly considered as a key novelty of this work.

Given the extremely slow rate of the motion (on the order of minutes) I can hardly believe that the low-temperature dynamics is dominated by quantum tunneling processes (as the attempt of using a WKB model would instead suggest). A noise-induced activation processes (as mentioned by the authors at some points) is a much more convincing explanation. Generally speaking, the discussion of the physical interpretation of the experiments is carried out in a quite obscure way in the manuscript, with the consequent risk of being misleading for the readers.

In parallel to the experiments, the temporal dynamics of the material system is simulated on an annealer machine. This is operated in a way to encode a transverse-field Ising model, which is a standard model where false vacuum decay processes are being studied in the theoretical literature. This model shares some general features with the electronic system and the annealer machine is subject to a similar $1/\nu$ noise spectrum, but (as the authors confirm at several points of the SM) the modeling protocol has no direct mathematical link with the physics of electrons in the material, in particular for what concerns the role of the kinetic energy term.

As another concern, I do not understand how information on the temporal quantum dynamics can be obtained using an annealer machine rather than a true quantum simulator. Unfortunately, the arguments offered by the authors (e.g. on the 'reverse annealing' feature) do not give much insight to justify this choice. If the authors had in mind to only focus on an incoherent relaxation dynamics (possibly with some activation by the noise), they should correspondingly and consistently amend the presentation to avoid misunderstandings. In this case, I am no longer sure that the wording 'false vacuum decay' would still be relevant; surely, 'macroscopic quantum tunneling' would no longer be relevant.

In summary, I am not convinced that the scientific content and the quality and clarity of the presentation are at the level required for publication on Nature Communications. As such, my recommendation is that the authors make a willing effort to improve the manuscript by clarifying all obscure points and removing all potentially misleading and/or overselling sentences and, then, resubmit the revised manuscript to some more specialized journal. When revising the manuscript, they should also take care that the main text is fully intelligible without having to continuously refer to the SM.

As minor issues:

-on pag.16 of the SM, I do not understand the sentence '... the domain walls cost zero energy in terms of interaction. However, the energy could be lowered still by getting rid of domain walls'. If the energy is not in interaction energy, where is it stored?

-I suspect that on pag.24 of the SM, '...them to the minimal possible value...' should be corrected into '...them to the maximal possible value...')

-The list of references may be reinforced with specific theoretical and experimental works on the relaxation dynamics of domain walls in different physical systems. This will facilitate the reader in assessing the points of novelty of this manuscript.

Reviewer #3 (Remarks to the Author):

In this article, the authors use two systems to show the metastable many-body states. One is the quantum material 1T-TaS₂, the other is an array of 2008 qubits in a programmable noisy superconducting quantum simulator. They claim that the simulations clearly display a crossover from T -activated dynamics to T -independent quantum domain melting that is experimentally observed in the STM measurements, and give remarkable insight into the mechanism for the emergence of false vacuum states in a microscopic system. I have a few comments on this article,

1) This article makes too much unrelated statements. For example, those statements, "Emergent metastability in non-equilibrium systems is a subject that touches everything from the origins of life..." "Understanding the dynamics of such systems is crucial for developing new quantum technologies..." are all beyond what can be claimed basing on their experiments.

2) Both a thermal activation process and a quantum tunneling process are involved in the reconfiguration process. But the quantum tunneling process is not well explained. For example, it is contradict to the classical Monte Carlo simulation. Why don't they try quantum Monte Carlo simulation? Or a quantum Monte Carlo in the presence of noise?

3) The energy landscape should be multiple dimension, why is it safe to use a simple one dimensional case.

4) How to obtain the reconfiguration rate from $f(t)$? It is unclearly explained.

5) The Fig1.d and Fig2.f are quite different. From Fig1,d we can clearly see the metastable state while in Fig2,f the metastable states are not clear.

6) How the $1/\nu$ noise affect the tunneling?

I don't recommend for publication in the present form. The above questions should be answered.

Reviewer #1 (Remarks to the Author):

In this manuscript, Jaka Vodeb et al. report a combined study of STM and quantum simulations about the time-evolution of the domain reconfiguration in 1T-TaS₂. The authors first prepare the domain structure in 1T-TaS₂ by the electrical pulse charge injection through STM tip. They record the domain structure periodically by STM and analyze the Hamming distance as a function of time. The authors further use the programmable noisy superconducting quantum simulator to simulate the reconfiguration dynamics. I am not convinced that simulating the domain reconfiguration dynamics in 1T-TaS₂ is important. I think this manuscript does not meet the high standard of Nature Communications. I do not recommend the publication of this manuscript in Nature Communications. Below are my detailed comments.

Author's response (AR): We thank the reviewer for his/her comments. We regret that the reviewer was not convinced of the importance of the simulations. In our response (and the didactically revised manuscript) we shall try and convince him/her otherwise. We also thank the reviewer for the detailed comments, which do not raise any major issues, and to which we respond below.

We first respond to the general statement of importance. (We note that in this regard the reviewer's opinion differs from the other reviewers.)

Simulations of low-temperature decay processes in 1T-TaS₂ are of interest for two main reasons:

- (1) for understanding the fundamental dynamics of a prototypical many-body quantum system with strong many-body correlations that displays topologically inhibited self-organized domain dynamics in 2D, and
- (2) for understanding the mechanisms that limit the performance of non-volatile charge configuration memristor devices based on 1T-TaS₂ (refs. 10,41,42) which display record speed and energy efficiency (ref. 10).

The domain reconfiguration is responsible for metastability, where topological effects inhibit classical dynamics (ref. 42) and quantum effects are those that are responsible for relaxation of the persistent state at low temperatures below 20K.

More generally, quantum simulation is an important contemporary field of study, which was recently demonstrated (also by this work) to be useful for addressing the problem of dynamics in open quantum systems, and noise-induced quantum dynamics in particular, which are beyond the reach of classical computers.

The processes that take place in 1T-TaS₂ are an excellent prototypical example for demonstrating and simulating emergent quantum decay with parallel experiment and simulation.

In summary, quantum simulations are a transformative tool, which we demonstrate here as a solution to solution to problems that surpass the capabilities of classical computers. From the practical viewpoint, we address an extremely important problem: origins of the decay of the non-volatile state in an ultraefficient memory device.

In the revised manuscript we have highlighted why simulations of domain dynamics are important.

Response

(1) What is the time interval for taking the STM topographies in Fig. 1c? Are the STM topographies in Fig. 1c taken on the same area?

AR: We thank the reviewer for pointing out the omission. (The information is implicitly given in Fig. 1 d, and also in the SI, but not explicitly stated in the main text, for which we apologise.) The images are of the same area. The time interval between images is 8 minutes.

(2) Since the author also report the temperature-dependent reconfiguration dynamics, in the main text, they should also show similar series of STM topographies as in Fig. 1c, but measured at different experimental temperatures.

AR: In fact, this is how Fig. 1e is obtained: by measuring the reconfiguration dynamics rates at different temperatures. Hundreds of such images were recorded at different temperatures. The actual series of images at different temperatures could be included, but this would require a presentation of more than 100 images, which we judged to be impractical. We considered that the one example shown in Fig. 1c is sufficient to present the concept, and some more pertinent examples are given in the SI. Perhaps the editor might suggest how to resolve this presentation issue.

(3) For the fixed temperature, does the reconfiguration rate change for difference areas?

AR: Indeed, the reconfiguration rate depends somewhat on the position. That's why in the supplementary information, in Fig.S10, at any given temperature, we have many different measurements in each interval. In the main text we show the average.

(4) In comparison with Fig. 1e, there are only a few data points in Fig. 1f. In order to make better comparison, the authors should measure more data points with the STM tip retracted.

AR: The purpose of the procedure of measurement with the tip retracted is to ascertain that the system is not affected by the presence of the measurement. It is not possible to add points if the tip is retracted. That is the point of the procedure.

We believe that the presented measurements unambiguously show that the tip does not significantly influence the rate of reconfiguration. This is confirmed by other controls and checks given in the SI.

(5) The authors perform the experiment with STM tip retracted to confirm that the noise in M does not originate from the STM tip. But it could originate from the electric field of the STM tip, because the electric field is long ranged. The authors should try to retract the tip, and reduce the bias voltage at the same time.

AR: In contact mode used for obtaining images, assuming a tip-to-sample distance of typically ~ 300 pm, so the electric field is $\sim 3 \times 10^8$ V/cm. When the tip is retracted to 50 μm , the field is ~ 200 V/cm. The difference is a factor of $\sim 10^6$. We conclude that if there is an electric field effect, then the 10^6 difference in field should have an effect in the rate, but we see that changing the field from 3×10^8 V/cm to 200 V/cm has no measurable effect on the relaxation rate. We conclude that the electric field cannot be responsible for the relaxation.

(6) Why the authors do not fit the last data point in Fig. 2g?

AR: The quantum hardware used to perform our quantum simulations does not allow us to sample from arbitrarily low temperatures. See next point for further clarification.

(7) Since they are several parameters in the simulation, I am not surprised to see that the authors can more or less fit the reconfiguration rate. The authors should provide stronger evidence that their quantum simulation is indeed applicable to the domain reconfiguration dynamics in 1T-TaS₂.

AR: The comment from the reviewer suggests that they believe we are experiencing an overfitting problem in our quantum simulation of the experiment. Since we cannot know for certain how the reviewer came to this conclusion, we assume that they misinterpreted the definition of our Hamiltonian used to describe the system of polarons in 1T-TaS₂. We assume that they understood that we fit all of the qubit couplers J_{ij} and qubit biases h_i to the experimentally obtained curve, which would give a valid concern for overfitting. However, as is stated in the main text, we map all of these parameters to a physical model of two-dimensional polarons with nearest neighbour repulsion. This automatically fixes all J_{ij} and h_i such that precisely this model is realized on the quantum simulator. The only TWO parameters that we are left with are the effective temperature of the system and the ratio $r=A/B$ of kinetic and potential energy amplitudes. The simulated curve is therefore not a simple fit to the data, but rather an emergent behaviour of the system, which coincides nicely with the data for only one value of $r=58$ as explained in the paper. We further clarify and emphasize this point in the revised text.

Reviewer #2 (Remarks to the Author):

The manuscript by Vodeb et al. reports a joint experimental and theoretical work on the relaxation dynamics of macroscopic domains in a 2d electronic crystal and its modeling using a quantum annealer machine.

The relaxation dynamics of complex many-body or quantum field systems brought out of equilibrium and trapped into metastable states is presently a very active field of research from both the theoretical and experimental point of view. This is a very interdisciplinary field of research that connects concepts of condensed matter physics and quantum dynamics and has deep implications even in cosmology for the so-called 'false vacuum decay'.

The present work specifically attacks the phenomenology of domain wall relaxation: in contrast to the traditional cosmological false vacuum decay where the decay process connects two spatially-homogeneous configurations differing by some macroscopic observable (aka two vacuum states), here the decay process consists of the motion of domain walls spatially separating different crystalline arrangements. As a common feature of the two phenomena, the temporal dynamics involves overtaking some potential energy barrier to exit from a metastable local minimum of the energy landscape. Whereas the physics of domain wall relaxation is -at least to my eyes- less intriguing and challenging than the traditional false vacuum decay, still the topic of the manuscript can be of some general interest for a wide community of physicists. In any case, I strongly recommend the authors to update the title of the manuscript and several parts of the text to faithfully match the actual content of the work and avoid misleading the readers.

AR: We thank the reviewer for acknowledging the importance of this work for a wide community of physicists. The comments have proved extremely valuable, and we have used them to motivate an extended rewriting of the manuscript, particularly the simulations, with significantly more attention to didactics that may arouse additional interest in the comparison of false vacuum decay (FVD) and quantum domain reconfigurations that lead to emergent

behaviour. The differences reveal some fundamental aspects of noise-assisted quantum processes in relation to FVD.

We agree with the reviewer that while there is a connection of domain wall relaxation with false vacuum decay, it is important to highlight the differences. We have done this in the substantially revised manuscript and updated the title to “Non-equilibrium quantum domain reconfiguration dynamics in a two-dimensional electronic crystal: experiments and quantum simulations”.

As a most interesting experimental observation, the dynamics of a complex, density-modulated electronic state is followed in real time by taking successive STM snapshots of the temporal evolution of the electron configuration after the system has been initialized in a domain state with a suitable current pulse. The main observation is that the relaxation follows an exponential law and its rate tends to a constant value at low temperatures and follows an thermally-activated-like behaviour at higher temperatures. No study of the dependence of the rate on other physical parameters is provided and only a generic interpretation of the activation energy value is provided. Such thermal activation behaviours are well-known in the literature on domain walls; also the specific material system under investigation, its electronic density-modulated states, and its STM imaging are well-established in the community, so they can be hardly considered as a key novelty of this work.

AR: We are glad to see that the reviewer finds our experimental observation ‘most interesting’. While thermally activated domain wall motion has been observed previously in magnetic systems and ferroelectrics for example, this is the first time it has been observed in a CDW system, and first time it has been directly measured in a time-resolved experiment by STM. We agree that T-activated behaviour is not the key novelty of this work. The cross-over to T-independent processes *is* novel.

Given the extremely slow rate of the motion (on the order of minutes) I can hardly believe that the low-temperature dynamics is dominated by quantum tunneling processes (as the attempt of using a WKB model would instead suggest). A noise-induced activation processes (as mentioned by the authors at some points) is a much more convincing explanation. Generally speaking, the discussion of the physical interpretation of the experiments is carried out in a quite obscure way in the manuscript, with the consequent risk of being misleading for the readers.

AR: This comment is very valuable, and as lead us to significantly rewrite the paper, with much more succinct description of the processes involved, with more careful definition of terminology. We acknowledge that the physical interpretation of the experiment might not have been explained as well as it could have been as is evident from the misinterpretation of the reviewer in that we try and explain the temperature independent processes with pure quantum tunnelling. This is in fact not the case, and we only use the WKB estimation as a probe for estimating the viability of quantum tunnelling as an explanation. We then elaborate that the presence of noise is KEY in describing the dynamics observed in 1T-TaS₂ and it is precisely for this reason that we make the choice of using a NOISY quantum simulator in order to reproduce the experimental dynamics. In light of this misunderstanding, we rewrote major parts of the text relating to the physical interpretation of data in order to clarify our point. We updated the abstract (and title) to make clear that we describe noise driven dynamics, and we dedicated 3 paragraphs in the new Discussion section to detailed explanation of noise driven dynamics. We also removed the WKB discussion in order to avoid misleading the reader.

In parallel to the experiments, the temporal dynamics of the material system is simulated on an annealer machine. This is operated in a way to encode a transverse-field Ising model, which is a standard model where false vacuum decay processes are being studied in the theoretical

literature. This model shares some general features with the electronic system and the annealer machine is subject to a similar $1/\nu$ noise spectrum, but (as the authors confirm at several points of the SM) the modeling protocol has no direct mathematical link with the physics of electrons in the material, in particular for what concerns the role of the kinetic energy term.

AR: We acknowledge the reviewer's concern that our transverse-field Ising model does not exhibit the same kinetic energy term as a typical electronic Hamiltonian would. However, as we argue in the revised text the model shares the same kinetic energy term in the limit of strong interactions. This is based on a paper we were not aware of before (Balducci, F., Gambassi, A., Lerose, A., Scardicchio, A. & Vanoni, C. Localization and Melting of Interfaces in the Two-Dimensional Quantum Ising Model. *Phys. Rev. Lett.* **129**, 120601 (2022)), where the authors show that domain walls in the transverse field Ising model behave as free particles in the background of strong interactions. This leads to an overall Hamiltonian that is the same as we use to describe polarons in 1T-TaS2 in the limit of strong interactions. We add and emphasize this point in the revised version of the manuscript.

As another concern, I do not understand how information on the temporal quantum dynamics can be obtained using an annealer machine rather than a true quantum simulator. Unfortunately, the arguments offered by the authors (e.g. on the 'reverse annealing' feature) do not give much insight to justify this choice. If the authors had in mind to only focus on an incoherent relaxation dynamics (possibly with some activation by the noise), they should correspondingly and consistently amend the presentation to avoid misunderstandings. In this case, I am no longer sure that the wording 'false vacuum decay' would still be relevant; surely, 'macroscopic quantum tunneling' would no longer be relevant.

AR: Indeed, our focus was specifically on incoherent relaxation dynamics and we have already agreed in a previous point to amend the title to “Non-equilibrium quantum domain reconfiguration dynamics in a two-dimensional electronic crystal: experiments and quantum simulations”, which moves us away from claiming that we study traditional false vacuum decay. However, the authors disagree that there are no quantum dynamics present even in the incoherent case. This is precisely why we decided to introduce the concept of incoherent macroscopic tunnelling (IMT) processes into the manuscript. There is a large body of literature dealing with the influence of various noise sources on quantum dynamics in the presence of an external bath. In the manuscript we cite for example: Leggett, A. J. *et al.* Dynamics of the dissipative two-state system. *Rev. Mod. Phys.* **59**, 1–85 (1987); Grabert, H., Olschowski, P. & Weiss, U. Quantum decay rates for dissipative systems at finite temperatures. *Phys. Rev. B* **36**, 1931–1951 (1987); Whiticar, A. M. *et al.* Probing flux and charge noise with macroscopic resonant tunneling. *Phys. Rev. B* **107**, 075412 (2023); and we add extra references

- Boixo, S., Smelyanskiy, V., Shabani, A. *et al.* Computational multiqubit tunnelling in programmable quantum annealers. *Nat Commun* **7**, 10327
- Boixo, S. *et al.* Evidence for quantum annealing with more than one hundred qubits. *Nat. Phys.* **10**, 218–224 (2014).
- Boixo, S., Albash, T., Spedalieri, F. M., Chancellor, N. & Lidar, D. A. Experimental signature of programmable quantum annealing. *Nat. Commun.* **4**, 1–8 (2013).
- Denchev, V. S. *et al.* What is the computational value of finite-range tunneling? *Phys. Rev. X* **6**, 1–19 (2016).

All of these references together show clearly that quantum tunnelling is possible (not just in theory but also in practice on the quantum annealer) in the presence of noise as well as temperature, which is also the main study of this work. The issue then becomes that of semantics. We assume that the referee has in mind pure coherent quantum tunnelling when the system is not coupled to a bath. We agree that this is not the case here. However, if a

system+bath *quantum* description is necessary to describe dynamics, we argue that incoherent quantum tunnelling is still applicable and indeed appropriate in our case.

The reverse annealing feature is used to introduce the kinetic energy term into the system, which then together with the interaction term causes IMT relaxation processes in the context of external noise and a bath. The best analogy we can use to help understand what the reverse annealing feature does is with an iterative measurement process, just as it is performed in experiment. The STM measures 1T-TaS₂ iteratively with a certain time step. What happens in between measurements is subject to noisy, thermal, and quantum dynamics of the system. Reverse annealing, on the other hand, emulates this process by iteratively turning on and off the kinetic energy term and measuring the qubit configuration at every »turned off« point. We apologize for not explaining this point well enough and make our best effort to do so in the revised version. We have also made an effort to explain what we mean by IMT processes in the revised version.

In summary, I am not convinced that the scientific content and the quality and clarity of the presentation are at the level required for publication on Nature Communications. As such, my recommendation is that the authors make a willing effort to improve the manuscript by clarifying all obscure points and removing all potentially misleading and/or overselling sentences and, then, resubmit the revised manuscript to some more specialized journal. When revising the manuscript, they should also take care that the main text is fully intelligible without having to continuously refer to the SM.

AR: We thank the reviewer for their constructive criticism, which helped to greatly improve the revised manuscript version. We hope that we have addressed all of the concerns of the reviewer and that they find the revised version suitable for publication.

As minor issues:

-on pag.16 of the SM, I do not understand the sentence '... the domain walls cost zero energy in terms of interaction. However, the energy could be lowered still by getting rid of domain walls'. If the energy is not in interaction energy, where is it stored?

AR: This is a good question. The energy is stored in the chemical potential term. If the domain walls were gone, the interaction energy would still be the same, because the repulsion is only between nearest neighbours, while the distances between polarons on each side of the domain wall is more than that. However, by introducing more polarons into the system, we could lower the contribution from the chemical potential, thereby lowering the energy of the whole system. The domain walls turn out to be difficult to get rid of in a Monte Carlo algorithm because a large number of polarons would have to be introduced in precisely the right locations within the domain wall, which is a highly unlikely event. We elaborated this point in the revised SI.

-I suspect that on pag.24 of the SM, '...them to the minimal possible value...' should be corrected into '...them to the maximal possible value...')

AR: No, that is incorrect. The couplings need to be ferromagnetic (negative values) in order for physical qubits to tend to assume the same value. The value is set to be -1 in our case. We thank the reviewer for the comment and add this point to the SM to avoid further misunderstandings.

-The list of references may be reinforced with specific theoretical and experimental works on the relaxation dynamics of domain walls in different physical systems. This will facilitate the reader in assessing the points of novelty of this manuscript.

AR: We have added appropriate references to the manuscript.

Reviewer #3 (Remarks to the Author):

In this article, the authors use two systems to show the metastable many-body states. One is the quantum material 1T-TaS₂, the other is an array of 2008 qubits in a programmable noisy superconducting quantum simulator. They claim that the simulations clearly display a crossover from T -activated dynamics to T -independent quantum domain melting that is experimentally observed in the STM measurements, and give remarkable insight into the mechanism for the emergence of false vacuum states in a microscopic system. I have a few comments on this article,

1) This article makes too much unrelated statements. For example, those statements, “Emergent metastability in non-equilibrium systems is a subject that touches everything from the origins of life...” “Understanding the dynamics of such systems is crucial for developing new quantum technologies...” are all beyond what can be claimed basing on their experiments.

AR: We thank the reviewer for the comments. While the statements hold we concede that without further explanation they may appear unrelated. We have revised the MS to revise such statements that may appear unrelated or vague, such as the ones mentioned. By the time the Conclusions are reached, the relation to the topics mentioned in the beginning become clear (to the reader who persists to read the Conclusions).

2) Both a thermal activation process and a quantum tunneling process are involved in the reconfiguration process. But the quantum tunneling process is not well explained. For example, it is contradict to the classical Monte Carlo simulation. Why don't they try quantum Monte Carlo simulation? Or a quantum Monte Carlo in the presence of noise?

AR: We apologize for the lack of clarity and make our best effort to better explain the quantum tunnelling process in the revised text. What we mean by quantum tunnelling or, more specific to our text, incoherent macroscopic tunnelling processes, are large reconfiguration events of the system of polarons in 1T-TaS₂ or qubits on the quantum annealer, enabled by quantum dynamics in the presence of noise and a bath. The dynamics would not be possible if not for the presence of both the interaction and kinetic energy term in the system's Hamiltonian, which do not commute with each other. The comparison with classical Monte Carlo was done on purpose, to show that if we only keep the interaction term and a bath, the temperature independent dynamics would not be possible in 1T-TaS₂.

The use of quantum Monte Carlo (QMC) is a very good suggestion and may be explored in a future study. Unfortunately, it falls outside the scope of this work because it is still an open question whether it would even be possible to perform such a non-equilibrium QMC simulation. One approach would be to simply cut off the sampling in the algorithm and check whether the distribution of states really does correspond to the experimentally observed one. We could also implement different kinetic energy terms in order to emulate the system of polarons more accurately. However, we want to emphasize that we are NOT sampling from an equilibrium distribution, which is what QMC is designed to do. In both 1T-TaS₂ and the quantum annealer, we are sampling the non-equilibrium trajectory towards equilibrium. In fact, if we would have waited for our system to equilibrate (in both 1T-TaS₂ and the quantum annealer), the measured T -independent rate of polaronic movement would simply always be 0. Furthermore, QMC sampling is done in an inherently unphysical manner, such that it can be efficiently implemented on classical computers. The quantum annealer, on the other hand, is actually performing the incoherent quantum dynamics that we are interested in probing.

3)The energy landscape should be multiple dimension, why is it safe to use a simple one dimensional case.

AR: A multidimensional version of the Kramers-Caldeira-Legget theory that we could apply to our experimental data currently does not exist. With this work, we hope to stimulate research into this direction. We agree with the reviewer that the assumption of safe usage of a onedimensional case may be misleading. We thank the reviewer for the remark and add this point in the revised text.

4)How to obtain the reconfiguration rate from $f(t)$? It is unclearly explained.

AR: We apologize for not clearly explaining the reconfiguration rate R . It is obtained from $f(t)$, which measures the fraction of polarons that moved from one STM image to the next, by simply dividing it with the time interval between two sequentially measured images. We added the explanation in the revised text.

5)The Fig1.d and Fig2.f are quite different. From Fig1,d we can clearly see the metastable state while in Fig2,f the metastable states are not clear.

AR: We agree that the presentation of metastable states should be made more clear for a fair comparison of both figures. Therefore, we added lines that indicate the location of metastable states during the relaxation sequence as well as an explanation in the figure caption.

6)How the $1/\nu$ noise affect the tunneling?

AR: This question suggests that we have not clearly explained the influence of $1/\nu$ noise on the system. In essence, the noise broadens any energy level that the system has. So, if we imagine an entire energy spectrum of, for example, a system of polarons in 1T-TaS₂, the energy gap that has to be bridged for a transition between different polaronic states to occur is effectively reduced by coupling to an external two-level system. The broadening of energy levels is given by the integral of the $1/\nu$ noise spectrum. For more details, we refer the reviewer to a paper that is cited in the revised version on the manuscript Amin, M. H. S., Averin, D. V. & Nesteroff, J. A. Decoherence in adiabatic quantum computation. *Phys. Rev. A* **79**, 022107 (2009). In the revised text we explain the effect of noise more clearly.

I don't recommend for publication in the present form. The above questions should be answered.

AR: We thank the reviewer for all the constructive comments and hope that we have adequately addressed them in the revised version of the manuscript.

REVIEWERS' COMMENTS

Reviewer #1 (Remarks to the Author):

The authors have addressed the comments in my previous review report, and I have no further comments.

Reviewer #2 (Remarks to the Author):

While the authors have made important improvements to the manuscript and have removed most of the questionable statements, I am still not fully convinced by several of the authors' arguments in reply to my first report. On top of that, I keep feeling that its scientific content is not at the level that is required for publication on Nature Physics.

Should the authors decide to resubmit the manuscript to some other, more specialized journal, I urge them to address my previous questions on the reason for using an annealer (rather than a quantum simulator) and, even more important, on the relation between the actual physical system to the transverse-field Ising model used in the simulations. I am in fact not satisfied by the arguments that the authors put forward in support of mapping the physical system onto the transverse-field Ising model.

Reviewer #4 (Remarks to the Author):

It seems as though the authors have significantly modified their manuscript to remove somewhat spurious claims addressed by reviewer 3. The authors address questions about Monte Carlo simulations as well as including a note in the manuscript on how Quantum Monte Carlo simulations with noise are not developed for non equilibrium systems. They mention in their response how QMC is designed to sample from an equilibrium distribution whereas the manuscript specifically studies non equilibrium dynamics. The authors address the need for a multidimensional version of the KLC theory in the manuscript and have added explanations for the reconfiguration rate as well as the effect $1/\nu$ noise has on the system.

In summary, I see the primary concerns of reviewer 3 as having been dutifully addressed.

**REVIEWER COMMENTS**

Reviewer #1 (Remarks to the Author):

The authors have addressed the comments in my previous review report, and I have no further comments.

Author's response (AR): We thank the reviewer for helping us improve our manuscript.

Reviewer #2 (Remarks to the Author):

While the authors have made important improvements to the manuscript and have removed most of the questionable statements, I am still not fully convinced by several of the authors' arguments in reply to my first report. On top of that, I keep feeling that its scientific content is not at the level that is required for publication on Nature Physics.

Should the authors decide to resubmit the manuscript to some other, more specialized journal, I urge them to address my previous questions on the reason for using an annealer (rather than a quantum simulator) and, even more important, on the relation between the actual physical system to the transverse-field Ising model used in the simulations. I am in fact not satisfied by the arguments that the authors put forward in support of mapping the physical system onto the transverse-field Ising model.

AR: We thank the reviewer for their constructive criticism, which did in fact help us to greatly improve our manuscript. In light of this, we provide a rebuttal to the final points made by the reviewer:

(i) The reason for using an annealer rather than a quantum simulator:

In our experiments we study polaronic domain reconfiguration events of tens of domains with domain sizes of several tens up to hundreds of polarons, where each polaron takes up to 13 atomic lattice sites. Even by reducing the requirement from 13 to 3 atoms per polaron in our theoretical model, a small single domain (~ 24 polarons) requires close to 100 atomic lattice sites ($\sim 24 \times 3 = 72$ atomic sites). Quantum simulators capable of simulating the Bose-Hubbard model have 80 atomic sites in total available at best [Chiu et al., 2019. String patterns in the doped Hubbard model. *Science*, 365(6450), pp.251-256.] which is simply not enough to create tens of domains with an average domain size of 100 polarons, required to emulate our experiments. The same argument holds for quantum simulation using digital quantum computers [Mi et al., 2024. Stable quantum-correlated many-body states through engineered dissipation. *Science*, 383(6689), pp.1332-1337.]

(ii) The relation between the actual system and the transverse field Ising model used in simulations:

The authors remain convinced that in the limit of a small kinetic compared to interaction energy, domain walls govern the dynamics of the extended transverse field Ising model and they exhibit a hopping-like kinetic energy term. The main reason is the observation of similar domain wall dynamics in both the experimental system 1T-TaS₂ and the quantum annealer, which is substantiated also theoretically by Balducci et al [Balducci, F., Gambassi, A., Lerose, A.,

Scardicchio, A. and Vanoni, C., 2022. Localization and melting of interfaces in the two-dimensional quantum ising model. *Physical Review Letters*, 129(12), p.120601.].

We reiterated these points in the revised manuscript.

Reviewer #4 (Remarks to the Author):

It seems as though the authors have significantly modified their manuscript to remove somewhat spurious claims addressed by reviewer 3. The authors address questions about Monte Carlo simulations as well as including a note in the manuscript on how Quantum Monte Carlo simulations with noise are not developed for non equilibrium systems. They mention in their response how QMC is designed to sample from an equilibrium distribution whereas the manuscript specifically studies non equilibrium dynamics. The authors address the need for a multidimensional version of the KLC theory in the manuscript and have added explanations for the reconfiguration rate as well as the effect $1/\nu$ noise has on the system. In summary, I see the primary concerns of reviewer 3 as having been dutifully addressed.

AR: We thank the reviewer for their time used in assessing our rewritten manuscript and are happy to see that they found the corrections satisfactory.